# Coccidioidomycosis in Immunocompromised at a Non-Endemic Referral Center in Mexico

**DOI:** 10.3390/jof10060429

**Published:** 2024-06-18

**Authors:** Carla M. Román-Montes, Lisset Seoane-Hernández, Rommel Flores-Miranda, Andrea Carolina Tello-Mercado, Andrea Rangel-Cordero, Rosa Areli Martínez-Gamboa, José Sifuentes-Osornio, Alfredo Ponce-de-León, Fernanda González-Lara

**Affiliations:** 1Infectious Diseases Department, Instituto Nacional de Ciencias Médicas y Nutrición Salvador Zubirán, Mexico City 14080, Mexico; carla.romanm@incmnsz.mx (C.M.R.-M.); andrea.tellom@incmnsz.mx (A.C.T.-M.); alfredo.poncedeleong@incmnsz.mx (A.P.-d.-L.); 2Clinical Microbiology Laboratory, Instituto Nacional de Ciencias Médicas y Nutrición Salvador Zubirán, Mexico City 14080, Mexicorosa.martinezg@incmnsz.mx (R.A.M.-G.); 3General Direction, Instituto Nacional de Ciencias Médicas y Nutrición Salvador Zubirán, Mexico City 14080, Mexico; jose.sifuenteso@incmnsz.mx

**Keywords:** *Coccidioides* spp., coccidioidomycosis, climate change, endemic mycosis, dimorphic fungi

## Abstract

The incidence and distribution of coccidioidomycosis are increasing. Information scarcity is evident in Mexico, particularly in non-endemic zones and specific populations. We compared the treatment and outcomes for patients with isolated pulmonary infections and those with disseminated coccidioidomycosis, including mortality rates within six weeks of diagnosis. Of 31 CM cases, 71% were male and 55% were disseminated. For 42% of patients, there was no evidence of having lived in or visited an endemic region. All patients had at least one comorbidity, and 58% had pharmacologic immunosuppressants. The general mortality rate was 30%; without differences between disseminated and localized disease. In our research, we describe a CM with a high frequency of disseminated disease without specific risk factors and non-significant mortality. Exposure to endemic regions was not found in a considerable number of subjects. We consider diverse reasons for why this may be, such as climate change or migration.

## 1. Introduction

Coccidioidomycosis (CM) is an endemic mycosis [1,2] that occurs in desert areas of the USA, near the Mexico–USA border, and in specific regions of Central and South America [3]. In Mexico, CM is endemic in the Pacific littoral and central zones; specifically, the states of Sonora, Chihuahua, Baja California, Southern Baja California, Sinaloa, Coahuila, Nuevo Leon, and Tamaulipas are recognized as endemic territories since epidemiological studies have established a prevalence of >30–60% from coccidioidin skin tests [3,4]. An endemic region has known environmental, climatic, and soil conditions supporting the saprobic form of *Coccidioides* spp. [5].

The Centers for Disease Control and Prevention (CDC) reported a 32% increase in CM cases between 2016 and 2018. In 2019, they reported CM cases in Arizona (144.1 cases per 100,000 people) and California (22.5 cases per 100,000 people) [6,7]. CM is not a reportable disease in Mexico since we may have underestimated the number of cases. Some data estimated that the burden of CM disease in Mexico is 7.6 cases per 100,000 people [8]. The rise in CM cases could be due to shifts in climatic conditions, as factors such as temperature, humidity, nutrient availability, and other soil microbes influence fungi. With climate change, we are facing a potential shift in the distribution of fungal diseases [9,10]. The burden of disease in Mexico is unknown despite its high prevalence because it has not been reportable since 1995, and access to mycological diagnosis is difficult [10,11]. Despite this, the global health community is realizing the importance of *Coccidioides* spp. with its inclusion in the fungal priority pathogens list developed by the World Health Organization (WHO) [12].

The gold standard of CM diagnosis is the culture of *Coccidioides* spp. from clinical samples and detecting spherules in tissue or cytology (mycologic criteria), although the culture has a low sensitivity. The histopathologic diagnosis with H&E staining shows the presence of spherules of various sizes (10 to 100 μm), thick-walled structures that contain hundreds of endospores (2 to 5 μm); spherules rupture when filled and release many endospores, each of which can spread to form a new spherule. The sensitivity for histopathologic detection of *Coccidioides* is 84%, and that for cytology is 75%. Serologic diagnosis can be used for CM with antibody or antigen detection; antibody test can be accomplished using complement-fixing antibodies (CF), expressed as a titer, or enzyme immunoassay (EIA) to detect IgM and IgG antibodies. Positive results with EIA commercial kits have a higher sensitivity (95%). In practice, however, most diagnoses of this infection are made based on positive serologies. As is true for many infections, ELISA results may be falsely positive for IgM due to rheumatoid factors or non-specific IgM, but a positive IgG indicates infection [13,14].

Up to 60% of primary infections do not show any clinical symptoms. When symptoms do occur, they can resemble flu-like symptoms or be mistaken for community-acquired pneumonia (CAP). The incubation period ranges from 7 to 21 days. Disseminated coccidioidomycosis (DCM) is most common in individuals with immunosuppression [15], pregnant women, and those more than 65 years old; increased risk has also been inferred among those who identify as African–American or Filipino [14,16]. The skin, lymph nodes, musculoskeletal system, and central nervous system (CNS) are the most common dissemination sites, occurring in less than 1% of cases [15].

We aimed to describe CM’s clinical characteristics and outcomes in a tertiary center in Mexico City (non-endemic medical center).

## 2. Materials and Methods

Data and Study Population. We performed a retrospective analysis of CM cases in adults between 2002 and 2022 in a tertiary care center in Mexico City, which is not a region endemic for *Coccidioides*. Demographic, clinical, and microbiological data, antifungal treatment, and outcomes were obtained from the electronic medical record. We compared localized CM vs. disseminated CM.

Definitions. We defined proven cases as those where spherules were identified in biopsy or cytology specimens or *Coccidioides* spp. was grown, while we defined probable cases as those where the diagnosis was based only on serology and a compatible clinical presentation. A disseminated infection was defined as clinical evidence of pulmonary disease plus one site of extrapulmonary dissemination of *Coccidioides* spp. to the skin, bones, central nervous system (CNS), blood, or other [17]. Immunosuppression was defined as prior or current use of steroids with a prednisone equivalent of 20 mg over two weeks or immunosuppressors for solid organ transplant (SOT), hematologic stem cell transplant (HSCT), rheumatic or autoimmune diseases, cancer chemotherapy, or HIV infection with <200 CD4 cells/mL. Malnutrition was considered in patients with a body mass index (BMI) < 18.5 kg/m^2^. 

Local cases were defined as those with known exposure in endemic areas in Mexico. Imported cases were those where probable exposure occurred in endemic areas outside Mexico. We considered endemic regions as those with known environmental, climatic, and soil conditions that support the saprobic form of *Coccidioides* spp. with reports of infection in these areas [4,6]. We looked for exposure factors such as travel or activities in regions described as endemic, which were considered exposure.

Statistical analysis. Descriptive statistics were used. Patient characteristics were described using percentages for categorical variables and median values with interquartile range for continuous variables. Patients with disseminated or localized disease were compared in terms of characteristics and dead versus alive using Chi-square or Fisher’s exact test (if >20% of expected cell counts are less than 5, then use Fisher’s exact test) and the Mann–Whitney U test as appropriate. A *p*-value < 0.05 was considered statistically significant. For some, the missing data allowed was at most 10–20%. STATA V14.0, USA was used for analysis.

## 3. Results

A total of 31 patients with CM were included in the analysis. Of the 31 CM cases, 13 (42%) we considered local cases, 5 (16%) were imported, and 13 (42%) had no history of living in or visiting an endemic area. Figure 1 describes the presumed site of exposure for CM infection, and Table 1 shows the characteristics of the CM patients.

Regarding the clinical and general characteristics of the CM patients, we found that the median age was 41 (IQR 30–53) and 22 (71%) were male. The most frequent comorbidity was pharmacological immunosuppression in 18 (58%) patients, of which 9 were taking steroids (the median equivalent of prednisone = 58 mg/day (IQR 35–75)). Other conditions by frequency were lymphopenia in 12 (46%) patients, type 2 diabetes (T2DM) in 11 (35.5%), rheumatic/autoimmune diseases in 7 (23%), HIV infection in 6 (19%), of which 4/6 (67%) had <100 CD4+ cell count, and malnutrition in 5 (16%); 19 (61%) patients had two or more comorbidities at the same time.

There were 30 (97%) proven cases and 1 (3%) probable case following the EORTC/MSGERC criteria. Localized disease occurred in 14 (45%) cases, all pulmonary, while DCM occurred in 17 (55%) cases. The most frequent sites of dissemination were lymphatic in seven (23%), skin in six (19%), osteomyelitis in five (16%), CNS in five (16%), spleen in one (3%), pericardium in one (3%), and eye in one (3%). Clinical symptoms were reported in 23/26 (88.5%) cases: fever in 16 (61.5%), cough in 11 (43%), myalgias/arthralgias in 3 (11.5%), and headache in 2 (8%).

The culture-based diagnosis was noted in 6 (19%) cases (reported as *Coccidioides* spp.), histopathology in 18 (58%), and both methods in 6 (19%). A single case of disseminated CM had positive serology (IgG) (1/8, 12.5%). Chest computed tomography (CT) showed nodules and micronodules as the most common abnormality in 26/31 (84%) cases, consolidations in 14/31 (45%), pleural effusion in 9/31 (29%), and cavitation in 5/14 (16%). No significant differences existed between patients with LCM and DCM (Table 1).

Antifungal treatment was indicated in 26 (84%) cases: 12 of LCM and 14 of DCM. Among five untreated patients, four died before diagnosis (microbiologic or histopathologic results) and one was lost to follow-up. Antifungal treatment for LCM was deoxycholate amphotericin B (dAmB) plus an azole in 5/12 (42%) cases, azole alone therapy in 5/12 (42%), and dAmB alone in 2/12 (16%). Patients with DCM received dAmB plus an azole in 6/14 (43%) cases, azole alone in 6/14 (43%), and dAmB alone in 2/14 (14%). Follow-up data were available for 19 patients, of which 6 (32%) presented with relapse of CM, 3 (50%) with LCM, and 3 (50%) with DCM; the median time of relapse was 460 days (IQR 78–738).

We found a 30% (n = 8) mortality rate at both six weeks and one year from the diagnosis. Among the patients who died, 75% had disseminated disease (*p* = ns), 62.5% had lymphopenia (*p* = ns), and 50% did not receive antifungal treatment (*p* = 0.006). Data on mortality characteristics are presented in Table 2.

## 4. Discussion

In this retrospective study of CM cases, we did not find any significant characteristics between localized and disseminated CM or between living and dead cases. Interestingly, a large percentage of patients were apparently not infected in known endemic areas of Mexico or the United States.

In Mexico, more than 1500 cases of primary CM and 15 cases of disseminated disease are estimated annually, which is most likely an underestimation [4]. Sonora, Chihuahua, Baja California, Southern Baja California, Sinaloa, Coahuila, Nuevo Leon, and Tamaulipas are recognized as endemic territories since epidemiological studies have established a prevalence of >30–60% from coccidioidin tests [5]. The Pacific littoral and central zones (extending from the southeast to Michoacan) are also endemic [6]. Other southern states, such as Campeche, Quintana Roo, Morelos, and Oaxaca, have reported increased incidence, but it is still being determined if this is due to migration or changing epidemiology [4]. According to Gorris et al., if CM were a reportable disease, better surveillance could help us better understand it and its real burden [18].

We were able to identify exposure to endemic regions in 58% of patients in Mexico or the South of the USA. However, 42% were lacking an obvious source of exposure. It has been reported that the incidence of CM has risen significantly in the past decades with the expansion of environmental sites containing *Coccidioides* (soil, small animal burrows) and human migration [19,20]. Another possible explanation is the increased population exposure to fungi due to soil disturbance caused by construction, travel for work or pleasure, and certain hobbies (e.g., armadillo hunting, model airplane flying, and archaeological digging) [21]. Involving a more thorough analysis, another potential cause of this shift is the current global climate change. The burden of fungal infections is changing significantly, which may lead to some fungal infections losing their endemic status. Cases are anticipated to be more common in areas beyond the usual exposure locations [9,10,22]. 

Of note in this study was the high frequency of disseminated CM, which might be explained by the selection bias due to the nature of this tertiary-care center. Although all patients had comorbidity, not all were considered immunosuppressed since some cases had T2DM and malnutrition, but there is evidence that both conditions alter the immune response. The role of cellular immunosuppression in the lack of control of endemic mycoses is well described, with rates of disseminated disease of 30–50% compared to 1–3% among immunocompetent individuals [15]. Specific cellular immunodeficiencies such as an altered interferon-γ pathway or interleukin-12 receptor β1 deficiency confer a high risk of CM dissemination [15], even if we do not have more patient data. In our study, the most frequent underlying comorbidities in disseminated CM were pharmacological immunosuppression, mostly high-dose corticosteroids, which may impact the progression of a newly acquired CM or reactivate prior latent foci. Despite our hospital receiving many SOT, HSCT, cancer, autoimmune disease, and chronic disease patients, these groups were infrequently affected in some series [2,23]. Lymphopenia was a frequent finding, highlighting the alteration of the immune response. Some studies found <500 total cell lymphopenia in up to 22% of patients with a fungal infection [13]. Most cases of CM in people living with HIV infection occur in highly endemic areas, and the CD4 cell count determines presentation. However, all our patients had advanced HIV infection, and half presented with localized disease [24]. It is known that DM can alter the immune response. A third of cases had T2DM, although information on glycemic control was unavailable. Several immune mechanisms are affected in this condition, mainly innate and adaptive T-cell function responses. Impaired production of type 1 IFN and reduced TNF-a production have been described [25]. Patients with T2DM commonly present with more severe illness, cavitary lung disease, and relapse despite azole treatment [26,27]. Genetic predisposition for severe disease has been related to HLA class alleles (A-9, B-5) and ABO group B, which have a greater frequency among those with African, American, and Filipino ancestry. The risk for DCM among Hispanics, the most frequent ethnicity in our region, has been variable [15,26].

A mortality of 30% was found without differences between localized and disseminated CM; but delay in the diagnosis, since 50% of the dead patients did not receive treatment. This rate may be associated with all patients having a comorbidity and a high proportion being immunosuppressed. Even though statistical significance was not observed, previous reports have shown higher mortality rates in patients with disseminated disease, underlying risk factors, or immunosuppression, reaching as high as 70% [1].

High clinical suspicion is essential to diagnose CM, particularly in individuals who lack clinical findings or in places not considered endemic; a thorough interrogation of risk factors is critical [28]. A recent description of the natural history of DCM showed that 22% of cases had no symptoms [1]. This study found respiratory symptoms in 33% of LCM cases and 50% of DCM cases despite the high frequency of abnormal chest CT. Initial evaluation should consider serological testing and imaging of clinically suspicious areas. Although a low yield of antibody testing with “classical assays” is expected for CM [29], novel EIA assays such as the MVista^®^ *Coccidioides* quantitative antigen test or IMMY^®^ sōna *Coccidioides* Ab LFA may aid early diagnosis; the latter is a sensitive, specific, and rapid test for the qualitative detection of IgM and IgG antibodies [30]. However, outside endemic regions, serological or antigen tests are frequently unavailable. Only 25% of patients were tested in our study, and a single positive test result was found. In contrast, diagnosis was mainly based on culture and histopathology. Serology can improve diagnosis in hospitals without Biosafety level 3 (BLS-3) laboratories, which is necessary to manage *Coccidioides* culture [31].

We acknowledge some limitations, the main one being the sample size; this limitation can be related to the fact that we did not find variables associated with DCM. Since exposure or travel information was gathered from medical records, memory bias or a lack of information could have hindered the identification of the source of infection in many cases. A larger sample of the population is needed to determine the differences between localized and disseminated CM. Regions not typically seen as endemic should be considered to accurately assess the global disease burden, especially among immunocompromised individuals known as susceptible hosts, even if worse outcomes are not identified. One strength is that we could describe cases diagnosed outside the typical endemic regions, which supports the theory of the changing endemicity of coccidioidomycosis. 

## 5. Conclusions

We described patients with coccidioidomycosis who had high mortality rates and a high prevalence of other health conditions, such as pharmacological immunosuppression, though there was no proof of statistical significance with disseminated diseases or mortality. Despite the possible bias, many patients were not exposed to endemic regions. The rise in cases without precise classic exposure could indicate a shift in epidemiology possibly linked to climate change.

## Figures and Tables

**Figure 1 jof-10-00429-f001:**
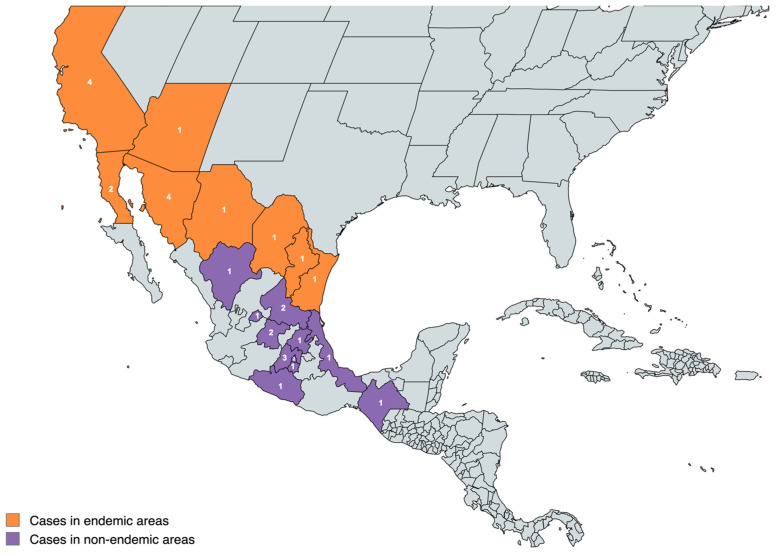
This map of Mexico and the Southwestern United States shows where infections are supposed to have been acquired. It is color-coded to differentiate between known or not known to be endemic regions [14].

**Table 1 jof-10-00429-t001:** General and clinical characteristics of patients with localized and disseminated coccidioidomycosis (CM).

Characteristics	All Cases N = 31 (%)	Localized CM N = 14 (%)	Disseminated CM N = 17 (%)	*p*-Value
Male sex	22 (71)	10 (71)	12 (71)	0.95
Age, median (IQR)	41 (30–53)	38.5 (29–48)	40 (30–55)	0.85
Live in endemic area	13 (42)	6 (43)	7 (41)	0.92
Imported cases	5 (16)	2 (14)	3 (18)	0.59
Comorbidities				
Type 2 diabetes	11 (35.5)	5 (36)	6 (35)	0.98
Rheumatic disease	7 (23)	3 (21)	4 (23.5)	0.88
Cirrhosis	4 (13)	2 (14)	2 (12)	0.62
HIV infection ^§^	6 (19)	3 (21)	3 (18)	0.79
Malnutrition	5 (16)	2 (14)	3 (18)	0.47
Lymphopenia	12/26 (46)	7/14 (50)	5/12 (42)	0.67
Baseline treatments				
Immunosuppressors	18 (58)	7 (50)	11 (65)	0.40
Corticosteroids	4 (29)	3 (21)	6 (35)	0.39
Clinical data				
Fever	16/26 (61.5)	8/12 (67)	8/14 (57)	0.61
Cough	11/26 (42)	4/12 (33)	7/14 (50)	0.39
Myalgia–arthralgia	3/26 (11.5)	1/12 (8)	2/14 (7)	0.56
Headache	2/26 (8)	1/12 (8)	1/14 (7)	0.72
Other symptoms	3/26 (11.5)	2/12 (17)	1/14 (7)	0.44
Culture alone	6 (19)	4 (29)	2 (12)	0.23
Histopathology	18 (58)	7 (50)	11 (65)	0.40
Both methods *	6 (19)	3 (21)	3 (18)	0.79
Outcomes				
Antifungal treatment	26 (84)	12 (86)	14 (82)	0.80
Relapse	6/31 (19)	2/14 (14)	4/17 (23.5)	0.51
Six-week mortality	8/27 (30)	2/12 (17)	6/15 (40)	0.18
One-year mortality	8/27 (30)	2/12 (17)	6/15 (40)	0.18

Key: IQR, interquartile range; ^§^ 4/6 (67%) HIV patients had <100 cell CD4+ count at diagnosis; * Both methods refer to culture plus histopathology. One case of disseminated CM was diagnosed with serology. A bivariate analysis between DCM and LCM did not show significant *p*-values.

**Table 2 jof-10-00429-t002:** Mortality at six weeks and one year of coccidioidomycosis (CM).

**Characteristics**	**All Cases** **N = 27 (%)**	**Dead** **N = 8 (%)**	**Alive** **N = 19 (%)**	** *p* ** **-Value**
Male sex	19 (70)	7 (87.5)	12 (63)	0.20
Age, median (IQR)	41 (30–55)	40.5 (31.5–44.5)	42 (30–58)	0.44
Comorbidities				
Type 2 diabetes	10 (37)	1 (12.5)	9 (47)	0.08
Rheumatic disease	7 (26)	1 (12.5)	6 (32)	0.30
Cirrhosis	3 (11)	2 (25)	1 (5)	0.13
HIV infection	5 (18.5)	3 (37.5)	2 (10.5)	0.09
Malnutrition	4 (15)	1 (12.5)	3 (16)	0.66
Immunosuppressors	17 (63)	4 (50)	13 (68.5)	0.36
Corticosteroids	8 (30)	2 (25)	6 (32)	0.73
Lymphopenia	12/26 (46)	5/8 (62.5)	7/18 (39)	0.26
Clinical data				
Disseminated	15 (56)	6 (75)	9 (47)	0.18
Culture diagnosis	6 (22)	2 (25)	4 (21)	0.82
Histopathology	14 (52)	3 (37.5)	11 (58)	0.33
Antifungal treatment				
Amphotericin B	3/22 (14)	1/4 (25)	2/18 (11)	0.88
Azole	10/22 (45.5)	1/4 (25)	9/18 (50)	0.08
Combination	8/22 (36)	1/4 (25)	7/18 (39)	0.20

Key: IQR, interquartile range; HIV, Human immunodeficiency virus.

## Data Availability

The original contributions presented in the study are included in the article, further inquiries can be directed to the corresponding author.

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
