# Peer review of "Coccidioidomycosis in Immunocompromised at a Non-Endemic Referral Center in Mexico"

_jof, 2024, doi:10.3390/jof10060429_

Round 1

Reviewer 1 Report

Comments and Suggestions for Authors

The authors present a retrospective descriptive study of clinical and epidemiological features of 31 coccidioidomycosis cases treated at a non-endemic referral center in Mexico. This topic is of great interest to the field providing more insights to the global distribution of coccidioidomycosis, especially in non-endemic regions. The manuscript would benefit from additional support of references. Additionally, while multiple statistical comparisons were described (X2, Fisher's exact test, Student's t or Mann-Whitney's U test), only bivariate p results were denoted in the table. Finally, if possible, I have suggested additional data collection that would strengthen the manuscript.

Major concerns:

Make sure endemicity is flushed out more in the introduction (L26-28). Information from the discussion (L165-167) could be included. I think it is important to preview which regions in Mexico are endemic and why. It may be helpful to visualize endemicity as a map panel or a bar graph with case counts by region (added to Figure 1).

Clarify L32 coccidioidomycosis is not reportable in Mexico. Are there any studies or statistics at all for the country? Any temporal studies? Add other citations about cases in non-endemic regions of Mexico (Fernández et al. 2017, https://doi.org/10.1016/j.riam.2017.03.006

L30-31 cite Gorris et al. 2019 (https://doi.org/10.1029/2019GH000209) and Salazar-Hamm & Torres-Cruz 2024 (https://doi.org/10.1007/s40588-024-00224-x) predicting CM impacted by climate change. There could be additional discussion of climate change predictions given this is important to the significance of this study.

Although not identified to species, there could be some discussion added about species variation in Mexico (Castanon-Olivares et al. 2007, https://doi.org/10.1196/annals.1406.047)

Minor concerns

L19 lowercase disseminated coccidioidomycosis

L21 define T2DM and reiterate this before mentioning in results (L123). I didn’t decipher this abbreviation until Table 2.

L40 italicize scientific name

L55 italicize scientific name

L64-66 definition of endemicity should be in the introduction

Figure 1 I think a monochromatic scale may be a better fit

L111 (Table) I am confused how there could be a higher number in “both diagnostics” than “culture” alone in disseminated CM column. This is likely an error. Double check table for additional mistakes.

L114 it is not clear what is meant by “Antifungal”. At first, I thought this was antifungal resistance, but I believe it means an antifungal was prescribed. Clarify.

L125 Please add more support to “Cases were proven…”

Comments on the Quality of English Language

Small changes in word choice were suggested to improve clarity in English. For example:

L81 (Table 1) Change “General” to “All cases”

L83 (Table 1) Change “man sex” to “Sex (male)”

L161 “In this retrospective series of CM...”  should change to “In this retrospective study of CM cases..”

Reviewer 2 Report

Comments and Suggestions for Authors

This manuscript describes a retrospective review of CM patients seen in a non-endemic region of Mexico. I believe the patients are all immune compromised, although that's not entirely clear. The authors attempted to make a lot of comparisons, ultimately finding that there were no differences between LCM/DCM and survival/death in this small cohort.

Title

The title doesn’t really fit the concept of the paper. The stated goal is to describe cocci in immune compromised individuals. There’s not much in the paper about the patients possibly infected outside of the known endemic area.

Abstract

Line 16: I’m not sure what “unique populations” is referring to.

Line 17: Does the use of immunosuppressants affect the disease or the disease severity?

Lines 17-21: I am not clear on the points being made here. It sounds repetitive, but I can’t tell for sure.

Introduction

Line 26: Is instead of are

Line 30-31: I think you’re trying to explain possible reasons for the increase, but it’s not clear.

Lines 35-37: This also isn’t clear and feels like it should be separated into a couple of sentences. More sensitive than what?

Lines 37-40: There should be a paragraph explaining spherules and endospores.

Lines 41-42: This is confusing. Your first point is about subclinical infection and then you go on to discuss a clinical picture in the same sentence.

Line 43: 1-3% of what? Infected people or the population or clinical cases?

Materials and Methods

Line 69: Dissemination or disseminated disease.

Results

Line 73: For clarity, these were all immunocompromised patients, correct? You’re comparing outcome in immunocompromised people, not comparing immunocompromised with immunocompetent? It would be good to clarify this.

Figure 1: The legend is very small and hard to read. I’d stick with the same format in the legend (i.e. “one case” rather than a case). The colors are hard to differentiate on the screen and it may be impossible for someone who is colorblind. I don’t count 31 cases from the map, although I may be misreading a color. If you could make the map more granular as far as suspected location of infection, it would be more meaningful.

Table 1: There should be a table legend. This is very hard to read as is. Could you please clarify what test you’re running (Fishers or Chi2 for example) for each. I’ve tried checking the math and on some of them I’m getting very different results. There should also be an explanation about why some are missing, i.e. fever, cough, headache. With this many comparisons you should adjust your p value as well.

Line 121: The table says 40 for median age.

Line 123: 58 mg per day or per dose?

Line 128: How were no clinical signs reported in 3? Why were the asymptomatic patients tested?

Line 140: 460 needs a unit

Line 143: I’d recommend including the actual p values.

Table 2: I am not getting the same results for all of these. From the ones I did get agreement, it looks like Chi2 was run to obtain the p value, but most of these would be more appropriate with a Fisher’s exact test. Please consider why you’re making a particular comparison. For example, relapse by definition only occurs in people that are alive. I’d also recommend a more complete legend for this table. All of the deaths occurred by 6 weeks (with 4 before diagnosis?). A one year survival comparison doesn’t add anything. You can state that all of the deaths were early.

Discussion

Lines 162-163: Nothing was significant.

Lines 169-170: Increased incidence compared to what?

Lines 172-174: This is a long run on sentence, and I’m not clear what you mean by “even in circumscribed areas” because wouldn’t that be where you’d expect to find it?

Line 175: explanation

Line 175-178: This is a couple of different points being made. I’m confused about better diagnostic methods since the introduction cites poor access to care and diagnostic tests.

Line 185: “have an impact on”

Lines 202-203: Run on sentence

Lines 207-209: Are you referring just to the antigen tests in this sentence? It’s not clear. Please check the name of Immy’s assay. If you expect Ab testing to be low yield for immunosuppressed patients, what is it about these tests that would be different?

Line 220: May present. Your results don’t support the statement as is.

Lines 223-224: If I’m understanding the chart correctly, there was no significant difference in mortality. Why might your mortality rate have been so high?

Conclusions

Your data don’t support your conclusions. You found no significant differences between DCM and LCM in this cohort. 

Comments on the Quality of English Language

There are a lot of run on sentences and the writing needs some clarity.

Reviewer 3 Report

Comments and Suggestions for Authors

Roman-Montes and colleagues describe Coccidioidomycosis in a non-endemic referral center within Mexico. The manuscript focused on individuals who did not visit or live in endemic areas but displayed other co-morbidities/ immunosuppressive treatments associated with CM. It is rather brief and could benefit from adding additional information before it is published. 

Major:

Title: Unfortunately, the title is rather vague, and I would argue that stating that this is present in Mexico would improve readership, as little is known about the impact of CM on the Mexican population.

Introduction: For disease impact, the authors can discuss that Coccidioides is listed under WHO's fungal priority list.

The CDC lists cases up to 2019, and Gorris et al. 2021 (DOI: 10.1175/wcas-d-20-0036.1) expect cases to continue to rise.

Results:

 It may be helpful to note if individuals who succumbed to CM had multiple comorbidities or were undergoing immunosuppressants at the same time. The current table makes it difficult to address this question.

Localized CM should be listed before disseminated CM.

Authors should discuss the classification of disseminated CM described in https://doi.org/10.1007/s40138-023-00274-3.

immunosuppression and Corticosteroids should be listed as treatments for both tables rather than comorbidities since there's an antifungal treatment section.

Discussion: justification to report CM in both North and South America (DOI: 10.3390/jof9010083)

Minor:

Figure1 : legend is rather small and unable to read # of cases 

italicize Coccidioides (ie line 40,55, ...)

lines 82-83 and 146 please rewrite as:  Sex: Male

line 86-87: traveled outside of endemic area

Comments on the Quality of English Language

The current manuscript needs minor edits on the table terminology such as

sex: Male

Overall the table legend needs to be consistent since the terms repeat between the 2 tables. 

Reviewer 4 Report

Comments and Suggestions for Authors

This paper from Carla Roman-Montes and collegues describes a small series of cases of coccidioidomycosis seen at The Instituto Nacional de Ciencias Médicas y Nutrición Salvador Zubirán in Mexico City, which is far from the known endemic areas for coccidioidomycosis in Mexico. I am pleased to see the interest in this infection and commend the authors for putting together this series. They divided their cases into localized Since it was based primarily on retrospective chart reviews, it’s not surprising that much information that they are missing some data would be of interest had the data been collected prospectively. In addition, because the number of cases that are included is relatively small (31 collected over a 20 year period) the study lacks statistical power to make meaningful comparisons between the two groups. 

I have several comments and suggestions for the authors:

1.     Please indicate what your referral area is, e.g., within Mexico City or a larger catchment area. 

2.     Were the patients referred because of the CM infection or for other reasons such as their underlying illnesses and the diagnosis of CM was made at your institution. 

3.      There were no cases of meningitis, which is a frequent presentation of DCM, I assume those patients would not have been referred to your hospital or is there another explanation. 

4.     What was the extent of the work-up to detect sites of dissemination? Did you have CF or equivalent immunodiffusion titers on patients? If so, did that result influence the extent of the work up for dissemination?

5.     On line 198 you start a brief discussion of ethnic and genetic predisposition to disseminated CM. I think “ethnic” susceptibility s the same as genetic susceptibility, but we have not yet learned the genes involved. The ABO types have not been conformed so are still speculative. You may want to add a reference to a recent paper by  Amt Hsu et al implicating mutations in (Cleck-7a (dectin-1) as a risk factor for dissemination.

6.     Reference 14 has the wrong authors. It also has no information about risk factors for dissemination other than pregnancy for women and low socio-economic status, probably a surrogate for increases exposure working as farm laborers. Were your patients migrant laborers? 

7.     I would recommend against including figure 1 as you don’t know how thoroughly travel/exposure information was sought and so few cases were attributed to any given region there really is no statistical basis for assigning different colors to those regions. 

8.     Although culturing for Coccidioides is a hazard for laboratory workers, it can happen inadvertently, for instance when sputum is being cultured for someone with cavitary lung disease. I agree that when you have pathology that shows spherules there is no need to also culture the sample. 

9.     In endemic areas, a positive IgG may not be enough to confirm the present illness and CM, but outside the endemic area that test is pretty good proof of recent infection. How available is serology in the region of Mexico from which your patients came?

10.  Antigen detection for Cocci is very insensitive and does have cross reactivity with Histoplasma, which is endemic in Mexico.

11.  Line 31. CM is not nationally reportable but is reportable in the states of California, New Mexico, and Arizona. Is it reportable in Mexico?

This paper from Carla Roman-Montes and collegues describes a small series of cases of coccidioidomycosis seen at The Instituto Nacional de Ciencias Médicas y Nutrición Salvador Zubirán in Mexico City, which is far from the known endemic areas for coccidioidomycosis in Mexico. I am pleased to see the interest in this infection and commend the authors for putting together this series. They divided their cases into localized Since it was based primarily on retrospective chart reviews, it’s not surprising that much information that they are missing some data would be of interest had the data been collected prospectively. In addition, because the number of cases that are included is relatively small (31 collected over a 20 year period) the study lacks statistical power to make meaningful comparisons between the two groups. 

I have several comments and suggestions for the authors:

1.     Please indicate what your referral area is, e.g., within Mexico City or a larger catchment area. 

2.     Were the patients referred because of the CM infection or for other reasons such as their underlying illnesses and the diagnosis of CM was made at your institution. 

3.      There were no cases of meningitis, which is a frequent presentation of DCM, I assume those patients would not have been referred to your hospital or is there another explanation. 

4.     What was the extent of the work-up to detect sites of dissemination? Did you have CF or equivalent immunodiffusion titers on patients? If so, did that result influence the extent of the work up for dissemination?

5.     On line 198 you start a brief discussion of ethnic and genetic predisposition to disseminated CM. I think “ethnic” susceptibility s the same as genetic susceptibility, but we have not yet learned the genes involved. The ABO types have not been conformed so are still speculative. You may want to add a reference to a recent paper by  Amt Hsu et al implicating mutations in (Cleck-7a (dectin-1) as a risk factor for dissemination.

6.     Reference 14 has the wrong authors. It also has no information about risk factors for dissemination other than pregnancy for women and low socio-economic status, probably a surrogate for increases exposure working as farm laborers. Were your patients migrant laborers? 

7.     I would recommend against including figure 1 as you don’t know how thoroughly travel/exposure information was sought and so few cases were attributed to any given region there really is no statistical basis for assigning different colors to those regions. 

8.     Although culturing for Coccidioides is a hazard for laboratory workers, it can happen inadvertently, for instance when sputum is being cultured for someone with cavitary lung disease. I agree that when you have pathology that shows spherules there is no need to also culture the sample. 

9.     In endemic areas, a positive IgG may not be enough to confirm the present illness and CM, but outside the endemic area that test is pretty good proof of recent infection. How available is serology in the region of Mexico from which your patients came?

10.  Antigen detection for Cocci is very insensitive and does have cross reactivity with Histoplasma, which is endemic in Mexico.

11.  Line 31. CM is not nationally reportable but is reportable in the states of California, New Mexico, and Arizona. Is it reportable in Mexico?

12.  In table 2, the abbreviation CM could mean Clinical Microbiology or coccidioidomycosis.

13.  Was lymphopenia separate risk factor or the result of either HIV or corticosteroid use? 12.  In table 2, the abbreviation CM could mean Clinical Microbiology or coccidioidomycosis.

13.  Was lymphopenia separate risk factor or the result of either HIV or corticosteroid use? 

Comments on the Quality of English Language

 Their English is good but sentence structure could be improved. 

Round 2

Reviewer 1 Report

Comments and Suggestions for Authors

Thank you for your resubmission. The additional detail strengthened the manuscript.  Figure 1 is greatly improved!

Minor suggestions:

L38: Replace total cases with the number of coccidioidomycosis with Arizona (144.1 cases per 100,000 people) and California (22.5 per 100,000 people) so it is easier to make comparisons to values from Mexico L40

L39-40: Revise for clarity, “In Mexico CM is not a reportable disease, thus numbers are likely an underestimation; however, a report suggests the burden of CM in Mexico is at least 7.6 cases per 100,000 people [7].”

L45-48: Revise for clarity, “Despite this, the global health community is realizing the importance of Coccidioides spp. with its inclusion in the fungal priority pathogens list developed by the World Health Organization (WHO) [11]. Remove the sentence, “The list is divided into three priority groups: critical, high, and medium. Coccidioides spp. is found in the medium priority group.”

L52 replace with “have a good sensitive” to “have a high sensitivity”

L93 remove extra “c”

L197-198: Revise for clarity, “A very interesting find was many cases do not appear to be exposed to known endemic zones.”

L207 Report as Gorris et al.

Comments on the Quality of English Language

Minor revisions should be made for clarity (see above).

Reviewer 2 Report

Comments and Suggestions for Authors

Thank you for the revisions and clarifications in the manuscript. I still think the title should reflect that these were only immunecompromised patients.

Comments on the Quality of English Language

There are a number of grammatical errors throughout the manuscript.

Author Response

Dear Reviewer,

Thank you very much for taking the time to review this manuscript. Please find the detailed responses below and the corresponding revisions/corrections highlighted/in track changes in the re-submitted file

Point-by-point response to Comments and Suggestions for Authors

Thank you for the revisions and clarifications in the manuscript. I still think the title should reflect that these were only immunocompromised patients.

Thank you for your comments; we decided to change the title. “Coccidioidomycosis in immunocompromised at a non-endemic referral center in Mexico”

Reviewer 4 Report

Comments and Suggestions for Authors

The authors have made an effort to respond to most of the original critiques, but there are several issues that remain, largely due to the small sample size. The reason to publish this study is as the authors say to raise the "profile" of this infection in Mexico by showing that there are cases presenting outside the known endemic areas for coccidioidomycosis in Mexico, and to raise the possibility that the infection may be acquired in areas of Mexico that are far from the known endemic areas. Here are specific suggestions for the authors. Sentence structure is sometimes awkward throughout the manuscript so I suggest it be edited by a native English speaker. 

The revised manuscript is improved but still has problems that should be addressed.

1.        L 18. Should read: We compared the treatment and outcomes for patients with isolated pulmonary infections and those with disseminated coccidioidomycosis, including mortality rates within 6 weeks of diagnosis.

2.        L. 35, reference 3. Is that referring to skin test or serology? 

3.        L 49. While it is correct that the isolation of the fungus from a patient is diagnostic if coccidioidomycosis, the detection of spherules by pathologists in tissue or cytology is equally pathognomonic because unlike yeast or hyphae, the spherule is unique to Coccidiodes. In practice however, most diagnoses of this infection are made based on positive serologies. As is true for most infections, there are false positive IgM results by ELISA due to rheumatoid factor but a positive IgG is indicative of infection. Please rewrite this paragraph.

4.        L 63. While that reference did estimate only 1% of cases had dissemination, there are numerous other studies that found a higher rate. The differences may be due to the genetics of the populations that were studied, as Arizona has a small African-American population and an even smaller Asian/ Pacific Islander population. Consider giving a range of estimates with references.

5.        L71. Consider “we defined proven cases as those where spherules we identified in biopsy or cytology specimens or Coccidioides was grown, and probable cases if the diagnosis was based only on serology and a compatible clinical presentation. Disseminated infection as defined as at least one extrapulmonary site of infection.” 

6.        The legend for figure 1 is not clear enough. I suggest “This map of Mexico and the Southwestern United States shows the places where infections were thought to have been acquired. It is color-coded to differentiate between known or not known to be endemic regions”. The map is in error are West Texas is a known endemic region in the United States, even though coccidioidomycosis is not reportable in Texas. If people were not asked about travel to that part of Texas that should be acknowledged as a limitation of this study.

7.        In table 1 do you have CD4 counts for the HIV infected patients? Dissemination is not expected until the counts fall to 2-300. You have a very low rate of dissemination. Was it the AIDS defining illness is any cases? This might be another message of your paper. The difference in the 6 week mortality rate is apparently not statistically higher in DCM but that could be due to short f/u and to relatively small numbers of patients. In the text you include a 1 year f/u. Why is that not in the table? Most of the Cocci-related mortality is due to overwhelming pneumonia and meningitis, and as a referral hospital you probably would not get any of the former cases.

8.        L 197. Suggest “An interesting finding was that a large percentage of patients apparently were not infected in known endemic areas of either Mexico or the United States.

9.        L 214-17. These sentences do not make sense. Suggest re-writing them to clarify your meaning. 

10.  Line 219. If you want to mention species, this should be modified since many environmental isolates from Baja California are very similar to those from the San Diego region and are C. immitis. Since there areas to CF titers.  no known differences in virulence or epidemiology, this is not likely to be of significance for your work and you may want to drop the paragraph. 

11. Reconsider your classification of limited infection since you did not have access to CF titers. His CF titers often prompt a search for clinically inapparent dissemination. 

Comments on the Quality of English Language

Their English is much better than my Spanish, but should be edited by a native English speaker. I made a few suggestions for improvement but did not have the patience to rewrite everything.

Round 3

Reviewer 4 Report

Comments and Suggestions for Authors

Line 188 change to "the median time to relapse was..."

Line 224 change to"In 46% of cases the there was no known exposure in an endemic area. "

Comments on the Quality of English Language

Much improved.